# Smart Farm Security by Combining IoT Sensor Network and Virtualized Mycelium Network

**DOI:** 10.3390/s23218689

**Published:** 2023-10-24

**Authors:** Nurdiansyah Sirimorok, Rio Mukhtarom Paweroi, Andi Arniaty Arsyad, Mario Köppen

**Affiliations:** Department of Computer Science and Systems Engineering (CSSE), Graduate School of Computer Science and Systems Engineering, Kyushu Institute of Technology, 680-4 Kawazu, Fukuoka 820-8502, Japan; paweroi.rio-mukhtarom223@mail.kyutech.jp (R.M.P.); arsyad.andi-arniaty825@mail.kyutech.jp (A.A.A.); mkoeppen@ieee.org (M.K.)

**Keywords:** smart farming, Internet-of-Things, mycelium, metaverse, multi-agent system, sensor network, artificial immune system

## Abstract

In today’s world, merging sensor-based security systems with contemporary principles has become crucial. As we witness the ever-growing number of interconnected devices in the Internet of Things (IoT), it is imperative to have robust and trustworthy security measures in place. In this paper, we examine the idea of virtualizing the communication infrastructure for smart farming in the context of IoT. Our approach utilizes a metaverse-based framework that mimics natural processes such as mycelium network growth communication with a security-concept-based srtificial immune system (AIS) and transaction models of a multi-agent system (MAS). The mycelium, a bridge that transfers nutrients from one plant to another, is an underground network (IoT below ground) that can interconnect multiple plants. Our objective is to study and simulate the mycelium’s behavior, which serves as an underground IoT, and we anticipate that the simulation results, supported by diverse aspects, can be a reference for future IoT network development. A proof of concept is presented, demonstrating the capabilities of such a virtualized network for dedicated sensor communication and easy reconfiguration for various needs.

## 1. Introduction

In the past decade, the Internet of Things (IoT) has become an important paradigm in the communication infrastructure as a generic structure to organize the collection, distribution, and evaluation of heterogeneous sensor data. It assigns a location, energy resources, and backbone connectivity to the sensors. Thus, it is also of importance for the design of sensor-based security infrastructures. Security and the IoT can be two-fold: IoT becomes the target of security (e.g., avoiding data manipulation and sensor damage) and the proxy for a real-world security state—for example, a sensor network to monitor forest fires or bridge stability. Many studies have been devoted to the first-tier, securing IoT networks. The most significant obstacles are security-related, including authorization, privacy, access control, system configuration, verification, management, and information storage, as highlighted by Khalid et al. [1]. Moreover, Cui et al. designed and implemented IoT infrastructure using sensor-based risk identification and processing the surveillance data to identify the severity of the fire event [2]. Although much research has been dedicated to securing IoT networks, less attention has been paid to the secondary tier. This is due to the traditional focus of IoT networks on collecting and forwarding sensor data to centralized cloud or service endpoints. However, the critical ’response’ aspect of IoT networks, where one sensor’s data can affect the behavior of others or modify the data collection and transmission process, is notably absent.

This paper focuses on the integration of multi-agent systems (MAS) and artificial immune systems (AIS) as cutting-edge models for the deployment of IoT security network infrastructures. These models boost system performance by promoting effectiveness, security, and adaptability. Utilizing this approach, IoT devices can work together, learn from one another, and independently respond to threats, ensuring optimal performance in dynamic environments. In addition, we propose the practical application of mycelium network growth in metaverse simulations, which has the potential to revolutionize real-world communication network structures. Through this groundbreaking method, we showcase how sensor responses facilitate interconnections in the physical world. By relaying information from real-world sensors in metaverse simulations through the IoT network, we conduct a thorough analysis and provide feedback. To ensure enhanced security and data distribution throughout the IoT network, we have implemented a security model based on an artificial immune system and a multi-agent system transaction model. This innovative approach holds particular significance in smart farming and could pave the way for significant advancements in this area. Our research objectives include achieving connectivity, data distribution, security transactions, and resilience, and we believe that combining these concepts and models in the metaverse dimension opens up a new framework for achieving these goals. The significant contributions of this paper are as follows:

This article is structured as follows. Section 2 outlines the related studies and the motivation behind the research. The proposed scheme is described in Section 3. Section 4 presents the materials and methods, followed by the results obtained in Section 5, the discussion in Section 6, and, ultimately, the conclusions drawn in Section 7.

## 2. Related Studies and Motivation

A mycelium is a fungus that grows anywhere, including underground, due to the characteristics in which it spreads to grow and find food so that it can reach plant roots as a food source. The following research proves the unique ability of the mycelium network. Suzanne explained in her studies that trees using mycelium networks are motivated by the need to secure their sources of carbon and to ensure the distribution and conveyance of information, which provides multiple opportunities for trees to take action to interact with their neighbors and adapt to the rapidly changing environment [3]. In another study in which Douglas fir trees were injured, the Douglas fir dumped its carbon into the network, and the ponderosa pine took it up; then, the defense enzymes of the Douglas fir and the ponderosa pine were ’upregulated’ in response to this injury [4]. The schematic of resources and signals documented to travel through mycorrhiza networks (a mycorrhiza is a symbiotic association between a fungus and a plant), as well as some of the stimuli that elicit the transfer of these molecules in donor and receiver plants, was elaborated [5]. These indicated that the common mycorrhizal networks (CMNs) mediate plant communication through nutrient exchange and even enhance plant health and disease resistance, including the activation of various chemical components, such as enzymes, peroxidase, polyphenol oxidase, chitinase, etc. [6,7,8]. Another study considered pathogen-infected defense signals. This experiment used two tomato plants in one pot. Then, in the soil, the CMN grew, connecting both tomato roots. In the treatment conducted, one of the tomato plants was attacked by insects and then covered with a transparent plastic bag. The plant increased the activities of various chemical processes so that they could be transmitted to other plants through the CMN [9]. We will attempt to deploy this treatment in our simulation.

The security system was developed with an IoT approach using a multi-agent system by Meftah Z. The studies simulated the placement of sensors in the environment as an agent to detect the environmental state. When the sensor detects an abnormal environmental condition, the sensor will deliver information to the IoT network, which is received by a master agent that contains a troubleshooting server that operates based on the status of the environment. Then, the level or condition of the environment is commmunicated to the service agent, which can be linked to public services such as the police, civil protection, or hospitals as a response [10]. In the development and implementation of a training system for rehabilitation through a suite incorporating inertial measurement unit (IMU) sensors, it can be integrated with a system that allows its use in the web, mobile, and virtual reality (VR) domains. For the patient, as an agent, together with the utilization of the Unity3D engine, exercises can be added to the system remotely, accommodating different rehabilitation methods. This allows the system to adapt the activities to the advancement and performance of the patient [11].

The artificial immune system (AIS) has various algorithms that detect attacks. One of them is called the dendritic cell algorithm (DCA). Sahar Aldhaheri’s studies developed a Deep Learning Dendritic Cell Algorithm (DeepDCA) adopted from DCA and a Self-Normalizing Neural Network to classify IoT intrusion and minimize false alarm generation. Moreover, they automated and smoothed the signal extraction phase, improving the classification performance. The results indicated that DeepDCA performed well in detecting IoT attacks [12]. Other studies have performed experiments and tests on the wireless sensor network interaction dataset called WSN-DS, employing distributed artificial immune systems for known and unknown attack detection in IoT network systems. The system showed high efficiency in detecting unknown network attacks on the WSN [13].

Based on the gap that we show in Table 1, we built a new IoT framework that included self-protection from AIS and a strong transaction for each agent from the MAS, an aspect that has not yet been investigated in the related work. Moreover, none of the previous approaches considers networking between the sensors themselves. This limits the flexibility, control, and hard-coded connectivity in the physical infrastructure of the wiring, routing, and energy supply in the real world and the available options for the situational response of such systems. To tackle this issue, in the present report, we propose adding a virtualized network as a means of sensor communication. More specifically, we showcase a proof of concept based on the idea of virtualizing the communication infrastructure within the scope of IoT in smart farming based on the simulation of natural processes, mycelium network growth, and communication within a metaverse-based implementation.

In this context, we are developing a groundbreaking security system for the IoT network. Drawing inspiration from an underground mycelium network simulated in a virtual reality environment, we are confident that our innovative approach will significantly impact IoT security. Our comprehensive methodology involves a combination of cutting-edge techniques, which we believe can be adapted to a wide range of industries, including modern farming systems.

## 3. Proposed Scheme

Our scheme describes how the Internet of Things works. The web service will work as a link to all devices that are scattered as agents that can work together so that the information can be received faster. The user agent, as the system manager, processes the data derived from the sensor, and the action agent serves as the executor; it can be an alarm, firefighter, farmer, etc., but, in this case, we focus on the farmer or smart farming. The sensor placement in the real world and underground environment is built on virtual reality using the OpenSimulator Application, a platform to conduct simulations. The scheme’s concept is shown in Figure 1.

A real-world situation, including sensor placement, is set up in a metaverse environment. It also models the underground soil (where the soil is not visualized, but the plant roots, fungi mycelium, and nutrient distribution in a cave-like environment and from an underground perspective are considered). We also consider sensors that possess not only sensing capabilities but also the capacity for reconfiguration.

Reconfiguration here can be defined as follows:Changes in sensitivity parameters, making a sensor more or less sensitive to a certain physically observable;Selection of the physical observables to be sensed, especially as many systems can include multiple sensors at one place (commonly, temperature and humidity are sensed within the same device, for example), and for the sake of saving energy;Changes in the frequency of measurements; orSpecification of a sensed agent, e.g., in a ’chemical nose’, where only one agent can be sensed at a time.

Thus, the real-world sensors can be forwarded to a backbone in the traditional way and then fed into the metaverse simulation. After some time, the outcome of the simulation can be sent back to the real-world sensors via the backbone, thus reconfiguring the sensors in the above-listed ways.

OpenSimulator is an open-source, multi-platform, multi-user 3D application server. It can create a virtual environment (or world) accessed through various clients on multiple protocols. We use a provider-hosted OpenSimulator server environment (version 0.92, provider Zetaworlds) for the metaverse environment. OpenSimulator is also equipped with facilities that can absorb data from sensors installed around the crop in the real world so that they can monitor the plant environment and activity in real time. These facilities are called REST-APIs Services; this server, built into the web service, is written in PHP with a lightly weighted REST API to distribute the data from the sensor to the cloud computing and OpenSimulator-based virtual worlds [14]. As shown in Figure 2. this service concept has a crucial role in distributing data, besides communication between the real world and the virtual world, so that data processing can be performed in OpenSimulator and then returned as a response or warning to sensors or other machines.

The aspects promoting such an approach and differentiating it from existing ones are numerous and include the following.

The virtual networking communication means can be changed at runtime rather easily. There is no need to follow the classical packet-based protocols or set up hardware; any other method is feasible and free to explore—for example, template-based concepts of AIS, transaction models of MAS, or concepts borrowed from vehicular networking.The whole system is capable of retargeting a sensing focus and schedule. For example, an alarm state ’heat’ at one location can trigger higher sensitivity to temperature at other locations, with the coordinated sensing of a spreading fire source via a chain effect of reconfiguring more sensors, or it may be identified as a local situation only if there is no further indication of the same alert situation at other locations. In this sense, the system can be seen as self-organizing to some degree.Being metaverse-based, the network can be visually inspected by a ’farmer avatar’, and it can be interactively influenced (e.g., placing new nutrients to promote mycelium growth, taking measurements, directly adding or deleting components), feasibility tests can be performed that do not harm the external environment, and all activities can be shared among multiple jointly performing users.Virtualizations can run in parallel, allowing for ensemble evaluations and other methods of data science.There can also be insights into the biological processes underlying mycelium networks that give new directions for smart farming.

However, the main question then is whether we can have indeed such a mycelium virtual network capable of scheduling the sensing of real-world sensors and promoting the detection of alarm states. In this report, we will describe the setup of such an IoT virtualization network. This requires an examination of the details of a mycelium network and how it communicates and forwards states, thus also identifying the transaction agent needed to ensure the communication between plant roots and the mycelium. It also requires a suitabe means of growing such a network in a way that reflects the natural growth process, and a proof of concept of how an alarm state at one plant location can be forwarded to other plants to varying degrees, as well as the identification of the main parameters that influence this plant-to-plant communication. The mycelium networks virtually utilize the Monte Carlo method to determine the node position as the growth direction of the mycelium to reach the root plant so that a large network is formed, and this makes it easier to transfer data. We provide more details in the next section.

## 4. Materials and Methods

In this section, we describe the materials and methods that are used to develop a framework for IoT infrastructure related to security in simulation, starting by preparing the materials or devices that will be used in building the IoT infrastructure design based on the concept that we have formulated, including selecting software that is easy to use and can support the effectiveness of the performance of the methods that we apply in the simulation process.

### 4.1. Materials

To conduct the simulation, we create a new environment in the digital world that is called the metaverse; the application’s name is OpenSimulator Server (FireStorm as client viewer), version 0.6.4. OpenSimulator provides a suitable environment for the study and offers various frameworks, such as a server–client architecture, grid architecture, avatar-based control, concurrency, and scripting support. Within the so-called hypergrid linking the different server simulations around the world, it is possible to design an experimental framework to conduct simulations that can be tested, analyzed, and improved through multi-institutional collaboration [15]. Some simulation has been conducted using OpenSimulator, such as mechanical maintenance training for an aircraft engine (virtual world training for aircraft engine maintenance before real-world practice) [16], as well as educational activities such as a virtual laboratory of archaeology [17] and the implementation of expert system courses utilizing the free software OpenSim (version 6.6.14.69596)Revision of final proofread in virtual university campuses [18].

### 4.2. Methods

Based on the concept explained above, the experiment will be performed in a metaverse environment that utilizes an avatar as an electronic image (as in a video game) that represents and may be manipulated by a computer user to give actions such as creating an object, implementing scripts acting on the object.

#### 4.2.1. Fungi Network Growth Simulation

First, we developed a fungi network growth simulation system in a metaverse platform. We used the Monte Carlo method to simulate the growth of fungi underground. The Monte Carlo method is a method of solving numerical mathematical problems by random sampling. The result of the Monte Carlo simulation is obtained by repeated random sampling. The Monte Carlo method solves an issue by selecting a sample value from a large population [19].

The Monte Carlo method is used to model the next growth point of the mycelium, as shown in Figure 3. We assume that the previous position of the mycelium is $P_{0}$, and the current position of the tip of the mycelium is $P_{1}$. The possible next positions of the mycelium’s tip are $\left(P_{2.1},P_{2.2},\ldots ,P_{2.n}\right)$. All positions of $P_{2}$ are possible as the next position of the mycelium’s tip growth. We represent the population in a box-shaped space. The box space is located around the current position of the mycelium’s tip. Each position inside the box space is a possible position for the next point of growth. This part implements the Monte Carlo method to choose a random position from all possibilities of $P_{2}$.

Figure 3 notation:
$P_{0}$Mycelium previous position.$P_{1}$Mycelium current position.$P_{2}$Mycelium next position.$V_{1}$Vector line from $P_{0}$ to $P_{1}$.$V_{2}$Vector line from $P_{1}$ to $P_{2}$.$\left(P_{2.1},P_{2.2},\ldots ,P_{2.n}\right)$All possible positions of $P_{2}$.αMaximum angle of the next mycelium growth point.βAngle formed from $V_{1}$ and $V_{2}$.

In 3D space, the location of an object is defined as (x,y,z). $P_{0}$ and $P_{1}$ form a vector $V_{1}$. When a random position $P_{2}$ is chosen, a vector $V_{2}$ is also formed by $P_{1}$ and $P_{2}$. Both vectors $V_{1}$ and $V_{2}$ form an angle β degree. Then, β degree is compared to α degree, an angle parameter, to choose a point. If β>α, the $P_{2}$ position is outside of α. We repeat the process until condition β<α. If condition β<α is fulfilled, the $P_{2}$ position is chosen for the next growth position.

From all of the possibilities inside the box space, a point is chosen as the next position of the mycelium’s tip. There is a cone formed from an alpha-angled boundary from the last point of the mycelium’s tip. If the location point (x,y,z) is randomly chosen in the box space and inside this cone, then the point becomes the next position of the mycelium’s tip. This cone aims to ensure that the growth direction is not led to the opposite position.

#### 4.2.2. Experimental Plan

As shown in Figure 4, we build an environment with six plants placed on the ground about a hollow space to also visualize the roots and mycelium network below ground, to illustrate the exchange of nutrients or as an alarm (warning to other plants) through the underground network built by fungi and plant roots to form an overall state of mutual symbiosis. More specifically, as we show in Figure 5, we consider the addition of a transactor object as a link between the roots and mycelium, which acts as a bridge for nutrient exchange.

Each node, illustrated as yellow circles, will connect many nodes to share nutrients for mycelium growth;The nutrient is depicted as a red-brown circle placed on the sideline of the mycelium and node, and it is absorbed by the mycelium for growth, depending on the mycelia needed and also based on the nutrients available;The transactor node, which is denoted as a black circle, is tiny; these nodes will perform nutrient exchange honestly and safely according to the mycelium and plant root requirements;The mycelium and roots are described as lines with different colors: the mycelium is a black line while the root is a yellow-brown color. Between the mycelium and root, there is a transaction node as a bridge to transfer data, described as a trading place.

#### 4.2.3. Specific Experimental Plan

We developed a diagram based on the nutrient exchange behavior between fungi and plant roots, as shown in Figure 6. In this study, we used the Symbol-Meaning-Value (SMV) theory to better comprehend nutrient exchange between fungi and plant roots. The SMV is a concept of data transformation with trilevel thinking in terms of symbols representing raw data, knowledge representing the meaning of the data, and wisdom representing the wise utilization of the value of the knowledge [20].

Based on the information in Figure 6, we utilized SMV as a guide to form the stages of nutrient exchange, which we illustrate as data exchange. First, raw data as nutrients absorbed by the mycelium are then changed to a Symbol (S) and then transported using Meaning (M), which is a chemical process involving transportation to reach the plant roots. Plant roots receive a Value (V) to take action (response to the value received) and vice versa, transferring data from the plant roots to the mycelium with the same process [21].

In this study, we choose OpenSimulator as an experimental platform because it can provide a multi-user shared collaborative environment. Moreover, an easily edited/updated VR environment can consist of a more structural sequence of scenes [22]. As shown in the Table 2, we form an area that resembles an underground environment and include the activities that occur in this area; this is where the simulation is conducted, starting from the random dispersal of mycelium seeds by giving commands to the system that was built before. The mycelium seeds grow as well, randomly given a growth limit of up to 600 times, while trees are randomly allocated oxygen, as much as 1000–1500, respectively. Nutrient distribution is carried out with a limit of 80 for each experiment.

After the Monte-Carlo-based growth simulation is performed for the specified number of steps, we stimulate one of the trees to examine whether the attack can provide information to other trees. With this stimulation treatment, it is hoped that there will be a change in the value received on the alarm side, not only for the tree being attacked but also for the tree growing close to the tree being attacked. It shows that the warning from the attacked tree will be fundamental for the surrounding trees to prepare themselves for the next attack.

## 5. Results

Based on the scenario that we describe above, we conduct a six-time experiment. Each experimental result is depicted in a graph. Each experiment is performed in 10 min and produces data every minute. The six-time experiment’s results are described in more detail below.

### 5.1. Experiment 1

Figure 7 indicates that the distribution of data (values) occurs when the attack stimulation occurs in plant 1 and then transmits the data (provides a warning to other plants) to plants 2, 3, and 4, while plants 5 and 6 do not receive the data transmission.

For the data flow that occurs on the transactor described in Figure 8, by having 19 randomly formed transactors shown on the blue line with a strong data flow intensity, it indicates that there is an attack on plant 1 if compared to the flow of data on plants 2, 3, and 4, indicating that they received a warning from plant 1, while plants 5 and 6 did not receive an alarm.

### 5.2. Experiment 2

Figure 9 indicates that the distribution of data (values) occurs when the attack stimulation occurs in plant 2 and then transmits data (provides a warning to other plants) to plants 1, 3, 4, 5, and 6. In contrast, plants 5 and 6 both receive the data transmission as a low signal.

For the data flow that occurs on the transactor described in Figure 10, having seven randomly formed transactors shown on the orange line with a strong data flow intensity, it indicates that there is an attack on plant 2 if compared to the data flow on plants 1, 3, 4, 5, and 6, suggesting that they received an alarm from plant 2; however, plants 5 and 6 received an alert with a low signal.

### 5.3. Experiment 3

Figure 11 indicates that the distribution of data (values) occurs when the attack stimulation occurs in plant 3, and then transmits the data (provides a warning to other plants) to plants 4, 5, and 1, while plants 2 and 6 do not receive the data transmission.

For the data flow that occurs on the transactor described in Figure 12, with 16 transactors randomly formed on the gray line with a strong data flow intensity, it indicates an attack on plant 3 if compared to the data flow on plants 1, 2, 4, 5, and 6, suggesting that they received an alarm from plant 3, but plants 2, 5, and 6 received an alert with a low signal.

### 5.4. Experiment 4

Figure 13 indicates that the distribution of data (values) occurs when the attack stimulation occurs in plant 3 and then transmits data (provides a warning to other plants) to plants 1, 2, 3, 5, and 6. In contrast, plant 6 receives the data transmission with a low signal.

For the data flow that occurs in the transactor described in Figure 14, by having 10 randomly formed transactors visible on the light orange line with a strong data flow intensity, it indicates that there was an attack on plant 4 if compared to the data flow on plants 1, 2, 3, 5, and 6, indicating that they received an alarm from plant 4, but plants 5 and 6 received an alert with a low signal.

### 5.5. Experiment 5

Figure 15 indicates that the data distribution (values) occurs when the attack stimulation occurs in plant 5 and then transmits data (provides a warning to other plants) to plants 3 and 4. In contrast, plants 1, 2, and 6 do not receive the data transmission.

For the data flow that occurs in the transactor described in Figure 16, with six randomly formed transactors shown on the light blue line with a strong data flow intensity, it indicates that there is an attack on plant 5 if compared to the data flow on plants 1, 3, and 4, indicating that it received an alarm from plant 5, but plants 2 and 6 did not receive a warning. On the contrary, plants 1 and 4 received an alert with a low signal.

### 5.6. Experiment 6

Figure 17 indicates that the distribution of data (values) occurs when the attack stimulation occurs in plant 6 and then transmits the data (provides a warning to other plants) to plants 3, 4, and 5. In contrast, plants 1 and 2 do not receive the data transmission.

For the data flow that occurs in the transactor described in Figure 18, with seven transactors formed randomly on the light green line with a strong data flow intensity, it indicates that there is an attack on plant 6 compared to the data flow on the plants 3, 4, and 5, indicating that they received an alarm from plant 6, but plants 1 and 2 did not receive a warning. On the contrary, plants 3 and 4 received a low signal level alert.

## 6. Discussion

The growth of the mycelium underground is complicated to visualize directly. However, it becomes feasible with the help of virtual reality; thus, we built an environment that can visualize the underground environment transparently using the Opensimulator application, and we created several objects to illustrate the growth of the mycelium, such as oxygen objects, nutrients, tree roots, nodes (as connections), and mycelium. In virtualization simulations, we can see in more detail how the mycelium grows by spreading underground from one plant to another and shapes an extensive network, then ensuring the relationship between the plant roots and mycelium, where these networks are a bridge for nutrition distribution. Moreover, each plant can sense whether there is an abnormal state around them; the plant will communicate by forwarding this information to other plants as an alarm through the mycelium network that cooperates with the plant root.

In addition, the placement of nodes using the Monte Carlo method illustrated as a connector in the real world to shape a network becomes a vital consideration, where these networks describe an IoT network. This is worthy of being employed to build a more robust IoT infrastructure. Conversely, the role of the multi-agent system in our simulation proves that accurate transactions between plants are possible because they act as agents that can distribute data to each other. Regarding security, in these networks, each agent will detect an intrusion, with the capability to not only sense but also forward information to another agent. It then initiates a response to prepare self-protection, which is described as an artificial immune system model that can detect intrusions that come from various directions, allowing a brief period of time to prepare self-protection.

Combining the mycelium network concept with the MAS and AIS models successfully distributed data from plants that were attacked to surrounding plants as an alarm. This can be illustrated and implemented in the real world so that sensors with high sensitivity can work more effectively in distributing data to other sensors as an alarm when under attack. Another benefit seen in this simulation is the use of the metaverse in the OpenSimulator application, not only as a simulation environment but also as one entity or agent. This application can receive data from several devices (sensors). Then, various concepts can be used according to the requirements for the data processing to delve deeper into the meaning contained therein. As a result, this information can be sent back to other entities in response, such as fire departments, hospitals, police stations, etc.

## 7. Conclusions

With support from various aspects, the combination of multiple concepts, and the addition of a minor treatment in the simulation, we have introduced the mycelium network as the inspiration for a new concept in IoT infrastructure. Due to the unique behavior of the mycelium network, which can connect one tree to another through a wide network underground, illustrating an IoT network in the real world, supported by the simulation results, it can provide new directions for research, especially within the scope of IoT in smart farming as a form of control, sensing, alarms, etc., in facing various unpredictable states; thus, this concept will have a positive impact in the future for farmers, eliminating the need to consider different types of attacks. For instance, each sensor can perceive environmental changes and communicate with other sensors to exchange data about these environmental shifts. This collaboration provides vital information to empower farmers to act appropriately. The concept of mycelium networks can also be used in many fields as a reference for frameworks related to the IoT network. These include not only farming but also disaster mitigation, intelligent buildings, smart cities, etc.

Our future work will focus on establishing an extremely secure infrastructure for the Internet of Things. The framework provided by our simulation will serve as a guide to assist us in accomplishing this objective. We plan to deploy devices in the physical world, perform inspections, arrange them based on proximity, and utilize various techniques to evaluate their performance. This approach will allow us to construct an IoT infrastructure that is dependable and impervious to threats in the long term.

The limitations of of this study relate to the use of only a simulation to explore the concept and distribution of data from the real world to virtual reality; real-world implementation is considered as future work. Considering the various limitations, especially on agricultural land, the placement of sensors is still a major obstacle. We will conduct surveys and tests for future solutions before determining the agricultural land and sensor placement.

## Figures and Tables

**Figure 1 sensors-23-08689-f001:**
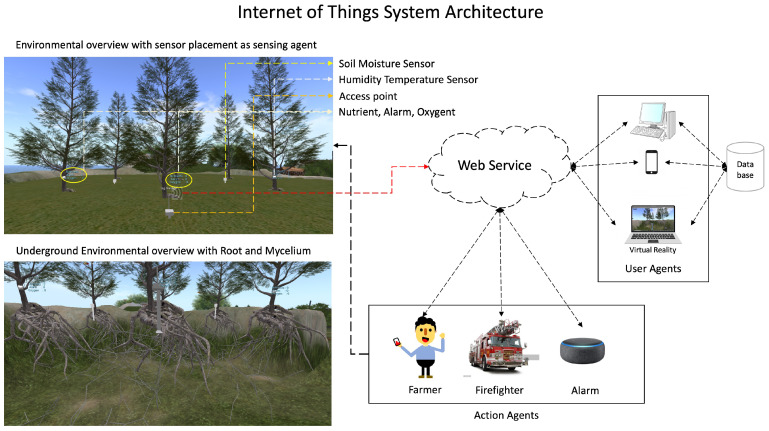
The proposed scheme.

**Figure 2 sensors-23-08689-f002:**
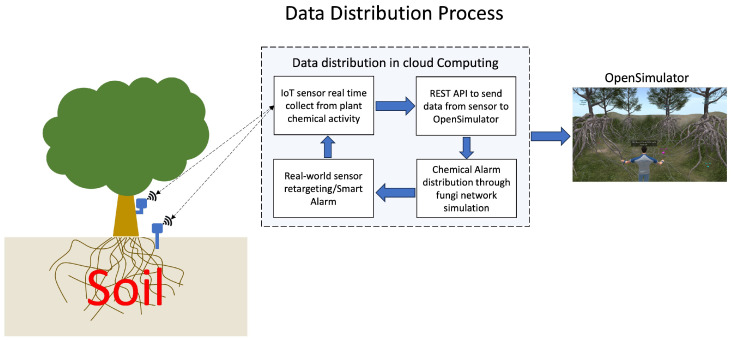
Description of data distribution from real-world sensor to OpenSimulator.

**Figure 3 sensors-23-08689-f003:**
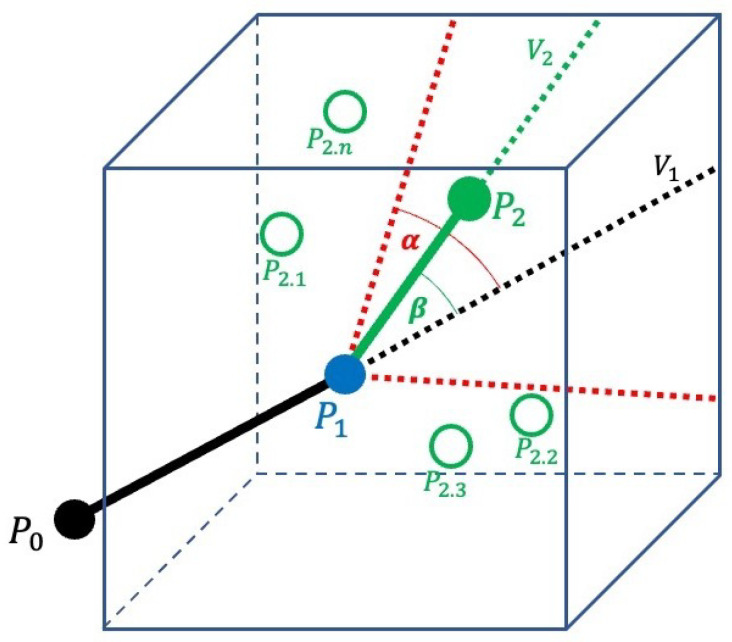
The Monte Carlo method in fungi network.

**Figure 4 sensors-23-08689-f004:**
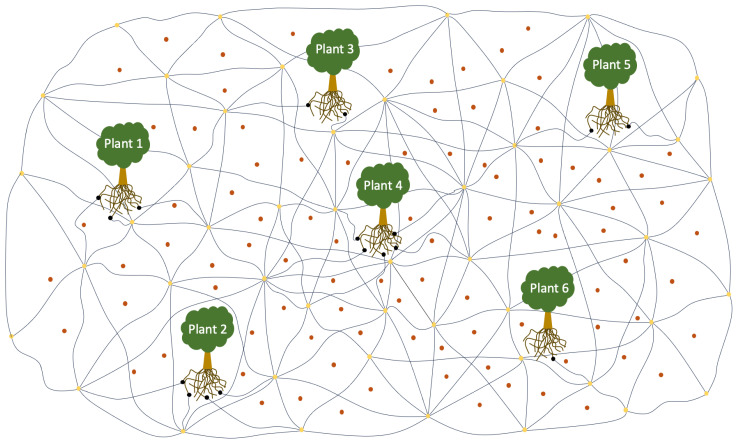
The experimental plan.

**Figure 5 sensors-23-08689-f005:**
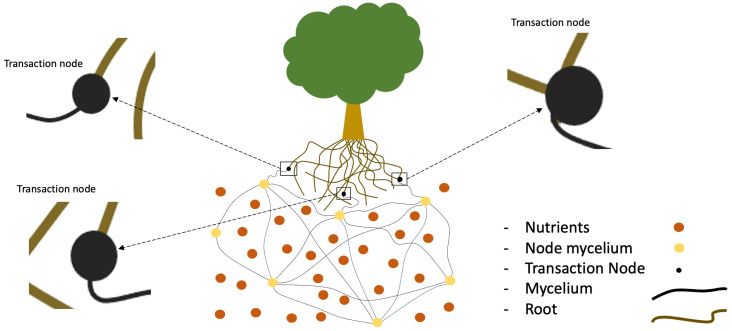
Detailed process connecting plant and root.

**Figure 6 sensors-23-08689-f006:**
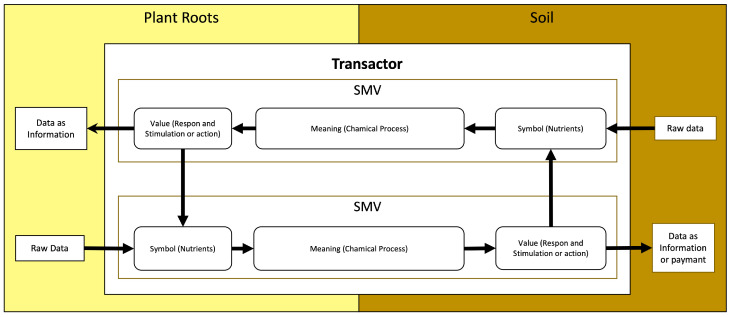
Description of the process of nutrient exchange.

**Figure 7 sensors-23-08689-f007:**
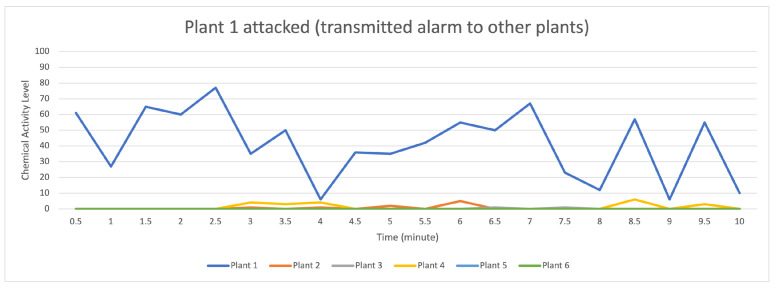
Description of chemical activity level on plant 1 when it is attacked.

**Figure 8 sensors-23-08689-f008:**
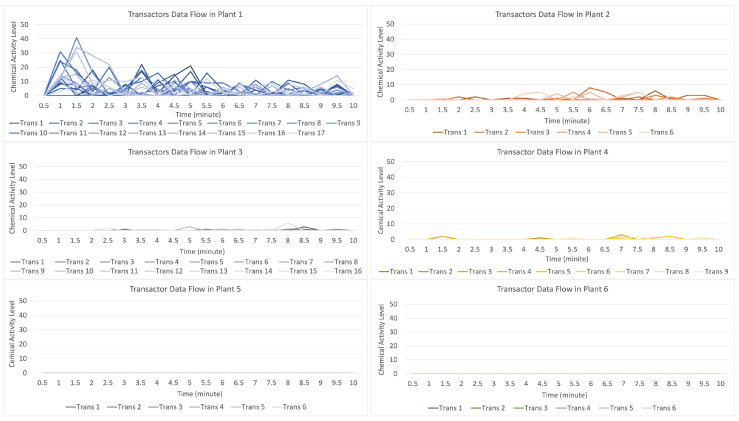
Description of chemical activity level on transactor when plant 1 is attacked.

**Figure 9 sensors-23-08689-f009:**
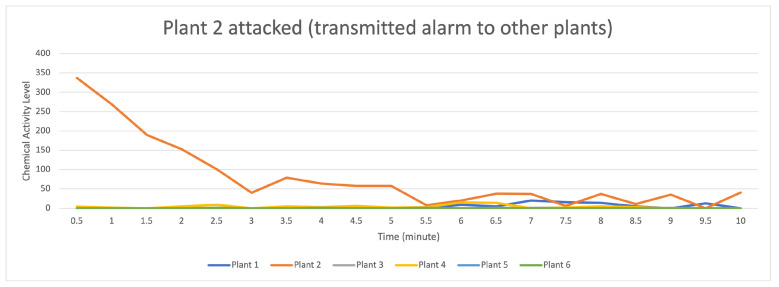
Description of chemical activity level on plant 2 when it is attacked.

**Figure 10 sensors-23-08689-f010:**
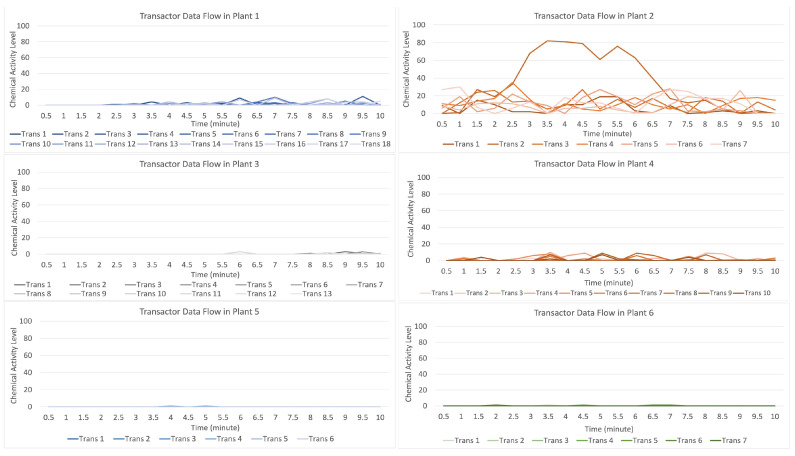
Description of chemical activity level on transactor when plant 2 is attacked.

**Figure 11 sensors-23-08689-f011:**
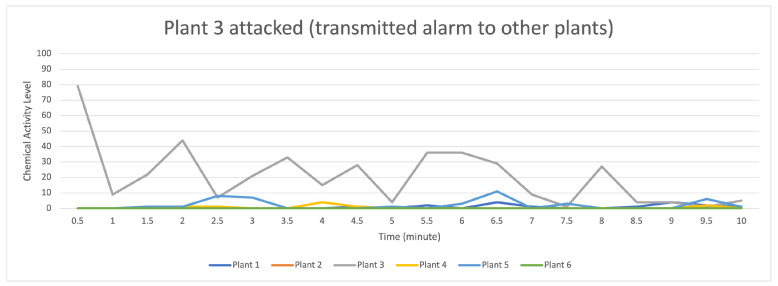
Description of chemical activity level on plant 3 when it is attacked.

**Figure 12 sensors-23-08689-f012:**
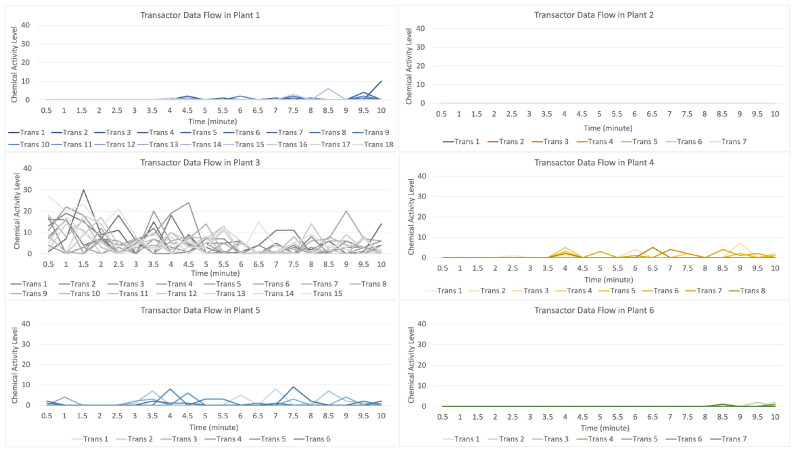
Description of chemical activity level on transactor when plant 3 is attacked.

**Figure 13 sensors-23-08689-f013:**
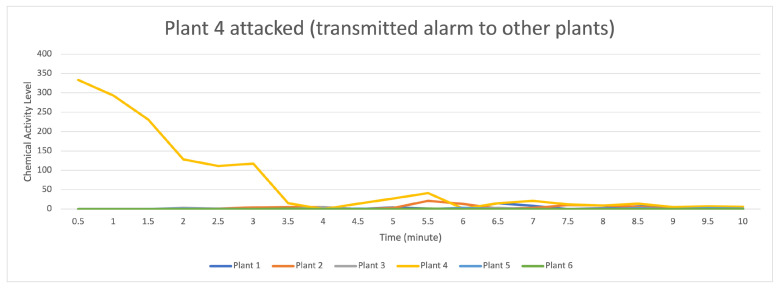
Description of chemical activity level on plant 4 when it is attacked.

**Figure 14 sensors-23-08689-f014:**
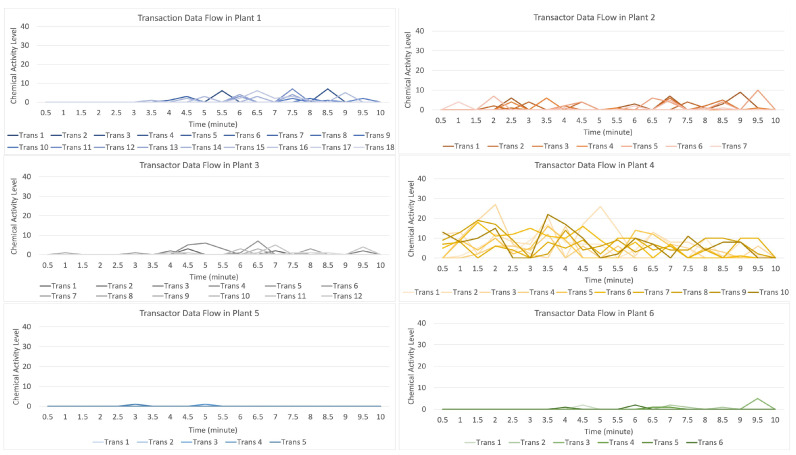
Description of chemical activity level on transactor when plant 4 is attacked.

**Figure 15 sensors-23-08689-f015:**
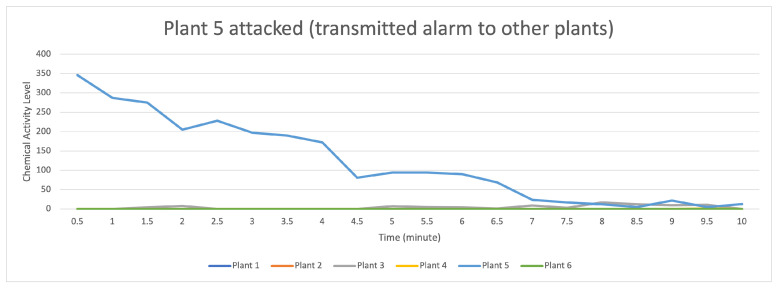
Description of chemical activity level on plant 5 when it is attacked.

**Figure 16 sensors-23-08689-f016:**
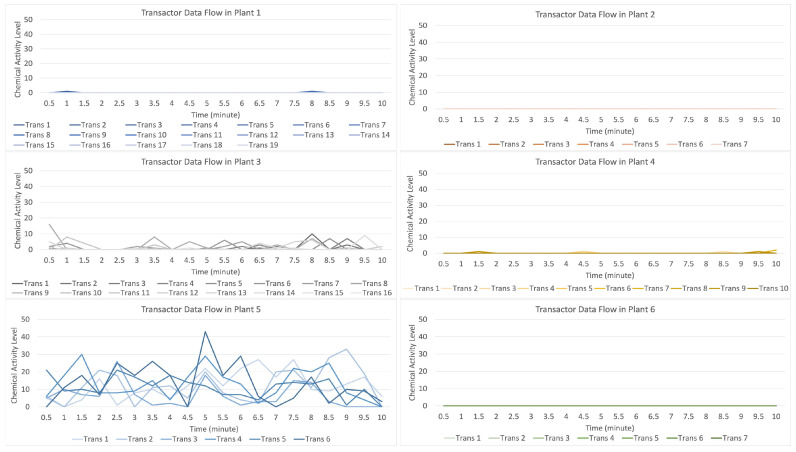
Description of chemical activity level on transactor when plant 5 is attacked.

**Figure 17 sensors-23-08689-f017:**
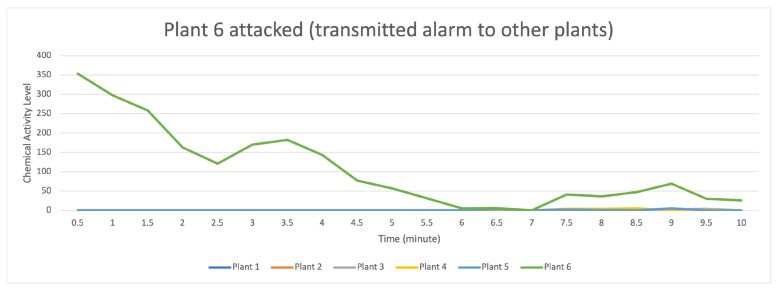
Description of chemical activity level on plant 6 when it is attacked.

**Figure 18 sensors-23-08689-f018:**
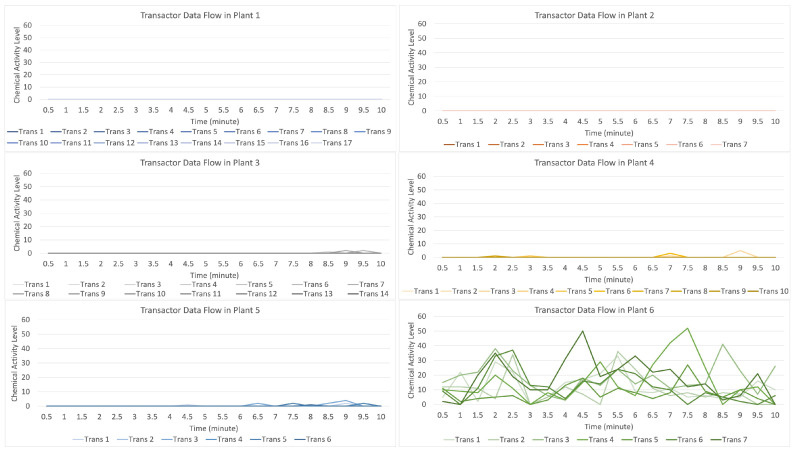
Description of chemical activity level on transactor when plant 6 is attacked.

**Table 1 sensors-23-08689-t001:** The differences in this study and previous studies.

Number	Previous Studies	This Study
1	Pathogen-infected defense signal delivered to another plant through underground common mycorrhiza network to prepare self-protection (real world as main idea) [9].	Attacked plant defense signal is delivered to another plant through an underground mycelium network to prepare self-protection (simulation, virtual world as main idea).
2	The sensor will deliver information to the IoT network, which is received by a master agent that contains a troubleshooting server operating based on the status of the environment [10].	In this study, each agent can receive information and is equipped with an intrusion detection system from the AIS model to prepare self-protection to solve problems to stop spread.
3	Utilize virtual reality that can control patients as an agent remotely [11].	Utilize virtual reality to sense environmental changes or detect abnormal states, particularly in smart farming.
4	Aldhaheri’s studies developed a deep learning scheme, namely a Self-Normalizing Neural Network, to classify IoT intrusions and minimize false alarm generation [12].	In our study, the artificial immune system for intrusion detection transmits an alarm to another agent to prepare for self-protection.
5	The wireless sensor network interaction called WSN-DS employs distributed artificial immune systems for known and unknown attack detection in IoT network systems in general [13].	This study employs artificial immune systems to detect intrusions in IoT network systems, particularly for smart farming, with sensor technology as a data collector.

**Table 2 sensors-23-08689-t002:** Scenario table.

Activity	Experiment 1	Experiment 2	Experiment 3	Experiment 4	Experiment 5	Experiment 6
Seed Dispersal	50	50	50	50	50	50
Restriction Growth	600	600	600	600	600	600
Oxygen Availability	1000	1000	1000	1000	1000	1000
Nutrition Availability	80	80	80	80	80	80
Attack Plants	Plant 1	Plant 2	Plant 3	Plant 4	Plant 5	Plant 6

## Data Availability

No new data were created or analyzed in this study. Data sharing is not applicable to this article.

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
