# Peer review of "Smart Farm Security by Combining IoT Sensor Network and Virtualized Mycelium Network"

_sensors, 2023, doi:10.3390/s23218689_

Round 1

Reviewer 1 Report

Dear Colleagues,

There is a lot of work invested in the article and the information is really valuable but I think should be solved some small issues before to be published

-          Usually keywords don't take (over) sequences from the title (eg. IoT, Security, Mycelium, Virtual and so on) – reformulate them in the way to reflect the article ideas and not just be redundant, please

-          Please specify the source of each figure / table (e.g. “Author's own processing” or other expressions / sources) between brackets after the name of the figure / table. Improve each figure / table (which is not “self- processing) with your “own contribution”

-          If you agree, in order not to abound with abbreviations / explanations in article’s body and for better identification, I recommend placing the explanation of all / each character(s) or abbreviation(s) (e.g. for each parameter, variable, attribute of and so on – and define each formula like equation, lema, theorem, proof of theorem and so on. The article must be easy to understand, both for specialists and for those less familiar with the subject. Please check the consistency and accuracy of each of them.

-          The “Literature Review”, (partially assimilated in your article in different chapters - including in Related Studies and Motivation chapter), should include in more detail the “gap” in existing literature / studies (meaning academic literature) and the innovative aspects brought by this paper (analysis for existing “literature” and the novelty and originality brought by this paper should be highlighted regarding “previous studies”) - please detail the gaps in the existing literature (like I said before - partially done in different chapters) and state more clearly / more explicitly the manner in which the article “addresses these gaps”.

-          I recommend that the "concrete" proposals with "practical" applicability and if possible... "measurable" be more clearly individualized (in a separate subsection / (sub)chapter of Results and Discussions, with direct mention at Conclusions). Actually, it would be interesting if the study would present some aspects more clearly related to the practical application of the study (examples) and its results (where could be applied, how could be applied and so on). Thus, please detail further the interpretation of the data analysis performed and its implications by reference to the scope of the research.

-          Please mentione (in a separate area after Conclusions chapter) more clear (the subjective and) limiting nature of the study (the limits of the research and the way in which these limits will be addressed in the future – if will be) and argue opinion regarding a possible modification of the investigation indicators also to reflecte and to have a holistic view on the topic

Reviewer 2 Report

This paper explored a metaverse-based framework inspired by mycelium networks for secure IoT communication in Smart Farming. It integrates an Artificial Immune System (AIS) for security and a Multi-Agent System (MAS) for efficient communication. The aim is to simulate mycelium behavior as an underground IoT network, providing insights for future IoT development. A proof-of-concept demonstrates its sensor communication capabilities and adaptability. The following suggestions aim to improve the overall clarity and presentation of the paper.

*The introduction section lacks clarity in conveying the contribution points and novelty of the study. It is essential to explicitly highlight these aspects in the introduction to provide readers with a clear understanding of the paper's unique contributions.

*While Section 2 provides a comprehensive overview of related studies, it would greatly enhance presentation if you could consider summarizing the key points and differences between this study and previous research in a table format. This would make it easier for readers to grasp the distinctions.

*The concept of a metaverse-based framework is intriguing. However, it needs further clarification. How does this framework differ from existing approaches in IoT communication? What are the key components and technologies involved?

* How exactly does the proposed framework mimic mycelium behavior, and how does this benefit IoT communication in agriculture?

*The text within Figure 1 appears unclear and needs improvement. Enhancing the legibility and clarity of the text within the figure is essential to ensure that readers can easily interpret its content.

*The quality of Figures 8, 10, 12, 14, 16, and 18 is suboptimal, making it challenging to discern the details they aim to convey. Please invest in improving the quality of these figures to enhance their readability and usefulness for readers.

*Carefully review the paper for grammar, syntax, and writing style. There are a few sentences that could be rephrased for clarity.

OK

Round 2

Reviewer 2 Report

The article has improved after revision. It can be accepted for publish

Ok